# Towards Sustainable and Nutritionally Enhanced Flatbreads from Sprouted Sorghum, Tapioca, and Cowpea Climate-Resilient Crops

**DOI:** 10.3390/foods12081638

**Published:** 2023-04-13

**Authors:** Mia Marchini, Maria Paciulli, Lorenza Broccardo, Maria Grazia Tuccio, Francesca Scazzina, Martina Cirlini, Eleonora Carini

**Affiliations:** 1Department of Food and Drug, University of Parma, Parco Area delle Scienze, 47/A, 43124 Parma, Italy; mia.marchini@unipr.it (M.M.); maria.paciulli@unipr.it (M.P.); francesca.scazzina@unipr.it (F.S.); martina.cirlini@unipr.it (M.C.); 2S-IN Soluzioni Informatiche, Limited Liability Company (Co., Ltd.), v. G. Ferrari 14, 36100 Vicenza, Italy; 3CUCI-University Center for the International Cooperation, University of Parma, Piazzale S. Francesco n, 3, 43121 Parma, Italy

**Keywords:** composite flours, sustainability, cowpea flour, sprouted sorghum, starch digestibility, micronutrient

## Abstract

This study aimed to develop high-quality flatbreads for low-income countries by using composite flours from climate-resilient crops, i.e., sprouted sorghum, tapioca, and cowpea, as partial alternatives to imported wheat. Through the experimental design, several flatbread prototypes were developed that maximized the content of sprouted sorghum and cowpea flours and minimized the content of wholewheat flour. Three of them were chosen based on the best textural, nutritional (highest intake of energy, proteins, and micronutrients—iron, zinc and vitamin A), and economic (cheapest in Sierra Leone, Tanzania, Burundi, and Togo) features. The physicochemical properties, in vitro starch digestibility, total phenolic content, antioxidant capacity, and sensory acceptability were also measured for the samples. The experimental flatbreads showed lower rapidly digestible starch and higher resistant starch contents than the control (100% wholewheat based), and were also richer in phenolic content and higher in antioxidant activity. Moreover, one of the prototypes was perceived to be as acceptable as the control for texture and flavour properties. The ranking test, performed after explaining the nature of the samples, revealed that the flatbread meeting the nutritional criteria was the preferred one. Overall, the use of composite flour from climate-resilient crops was proven to be an efficient strategy to obtain high-quality flatbread.

## 1. Introduction

Developing bakery products by using composite flours is currently a research topic of great interest, as it represents a way to improve the nutritional value of standard wheat-based products [1]. If the composite flours are obtained from climate-resilient, nutritious, and affordable crops, the approach can be particularly efficacious in developing countries. This approach has the advantage of saving money by reducing the import of wheat flour and providing a better supply of macro and micronutrients in the diet. Moreover, many more business opportunities along the value chain can be created by the use of local agriculture [1,2]. 

In the Middle East, North Africa, and Central Asia, bread has been a staple food for centuries, and its consumption is among the highest in the world, being consumed in almost every meal [1,3]. In these countries, the type of bread traditionally produced and consumed, especially at home, is flatbread. Flatbread may be obtained from flours with different gluten contents, which give rise to either a compact and elastic dough or to a semi-fluid batter [4]. One type of flatbread is unleavened flatbread, made from a flattened piece of dough containing flour, water, salt, and other optional ingredients [3]. For unleavened flatbread, the exceptionally high viscoelastic performance of wheat is not required. The use of composite flours for flatbread production, obtained from locally grown crops, should be actively encouraged to decrease the use of imported wheat while producing nutritionally enhanced bread [1,2,5,6]. 

Sorghum (*Sorghum bicolor* [L.] Moench) is a food crop cultivated in marginal lands in more than 100 countries. Grown primarily for food purposes by low-income farmers, it offers a staple for over 500 million poor and food-insecure people in around 30 countries in subtropical and semi-arid regions [7]. Therefore, it is potentially suitable for use in composite flours [7]. Sorghum sprouting is a sustainable technique that has been proven to improve its nutritional profile, although it worsens the starch functionality [7,8]. 

Cassava (*Manihot esculenta* Crantz) is a starchy root common among the lowland, sub-humid tropics of Africa; it is a primary source of calories for around two-fifths of all Africans [9]. Its high tolerance to drought and harsh climatic conditions, its high yield in poor soil, and its year-round availability make cassava a dependable crop for food security and various traditional food applications [9]. In addition, its flour (i.e., tapioca) exhibits excellent starch physicochemical and pasting properties, together with a mild non-cereal taste. Therefore, also cassava flour can partially replace wheat flour in bakery products, thereby reducing both expenditure on wheat imports in low-income countries and cassava post-harvest losses [10,11]. 

Cowpea (*Vigna unguiculata* L. Walpers), also known as black-eye pea, is one of the most important edible legumes contributing to food security and the maintenance of the environment for millions of small-scale farmers in developing countries [12,13]. Cowpea is a cheap source of protein and carbohydrate, as well as fibre and bioactive compounds [14]. To encourage higher consumption of this legume, recent efforts have increasingly focused on its use in composite flours to improve both the technological behaviour and nutritional profile of bakery products, revealing good functional properties of both protein and starch components [5,6,12]. 

In a previous study [15], we evaluated the technological properties of sustainable composite flours made of sprouted sorghum, tapioca, cowpea, and wheat flours at relative abundances of 50%, 33%, and 25% (*w*/*w*) flour basis. Principal component analysis showed that a 50% *w*/*w* sprouted sorghum and tapioca flour blend exhibited a technological profile similar to that of wheat flour, which may be of interest in the development of bakery products when the viscoelastic performance of gluten is not required. Additionally, a composite flour made from sprouted sorghum, tapioca, cowpea, and wheat flours was proven to be a compromise between flour’s technological and nutritional qualities, while reducing the use of imported wheat flour and cassava post-harvest losses [15]. Considering that the quality of a flatbread is related to the technological and nutritional properties of the used flours, along with their percentage in the blend, there is a need to optimize formulations to improve the overall quality of the finished products. As a result, the aim of this study was to develop nutritionally enhanced, economically sustainable, and texture acceptable flatbread formulations targeted to low-income countries (e.g., African countries). Formulations including composite flours made of sprouted sorghum, tapioca, cowpea, and wheat flours were characterized for their physico-chemical properties, nutritional traits, and sensory features. 

## 2. Materials and Methods

### 2.1. Flours 

Sorghum kernels were soaked (25 °C, 90% Relative Humidity—RH, 16 h), sprouted (25 °C, 90% RH, 72 h), dried (40 °C for 12 h), and milled (Labormill, BONA, Monza, Italy) to produce refined flour, middlings, and bran, which were then recombined as wholemeal sorghum flour (SS). The processing conditions used to obtain flour from sprouted sorghum were chosen based on previous experiments [16]. The sprouting conditions led to sorghum protein and starch hydrolysis, with a consequent increase in the total amino acids content, starch digestibility, and water- and oil-holding capacities [16]. The proximate composition (%, g/100 g d.b.) of the SS flour was: carbohydrates 83.6 ± 0.1%; proteins 12.0 ± 0.0%; fats 3.1 ± 0.0%; ashes 1.3 ± 0.0; and moisture 12.7 ± 0.0% wet basis (w. b.) [16]. Tapioca flour (T) (proximate composition: carbohydrates 99.3 ± 0.0%; proteins 0.6 ± 0.0%; fats 0.0 ± 0.0%; ashes 0.1 ± 0.0%; and moisture 11.3 ± 0.0%) and cowpea wholemeal (C) flour (proximate composition: carbohydrates 68.2 ± 0.1%; proteins 27.4 ± 0.0%; fats 1.4 ± 0.0%; ashes 3.1 ± 0.0%; and moisture 12.0 ± 1.0%) were purchased from Molino Bongiovanni S.r.l. (Villanova Mondovì, CN, Italy). Wholewheat flour (W) ((proximate composition: carbohydrates 81.8 ± 0.2%; proteins 14.8 ± 0.3%; fats 1.8 ± 0.0%; ashes 1.5 ± 0.0%; and moisture 10.7 ± 0.0%; rheological parameters: W = 240 J 10^−^^4^ and dough tenacity/dough extensibility P/L = 0.55; Italian legislation—(D.P.R., Presidential Decree 187, 2001)) was purchased from Molino Grassi S.p.A. (Fraore, PR, Italy). The technological functionality of the flours used was reported in a previous paper [15]. 

### 2.2. Flatbread Formulation, Production, and Characterization 

#### 2.2.1. Experimental Design

An experimental fractional factorial design (FFD) with Resolution V was used to investigate the effect of the SS, T, C, and W levels (input factors, independent variables) on the textural properties of the derived flatbread (responses, dependent variables). The experimental FFD was chosen for screening purposes to evaluate the preliminary significance of the variables and their interactions. Water level was included as an uncontrolled factor, as its use as a controlled/constant input produced a set of unfeasible experiments in preliminary trials, i.e., unworkable doughs due to too little or too much hydration. A centre point and two levels (minimum and maximum) for each input factor were selected in preliminary screening tests to meet different criteria. Firstly, the formulations had to include all four types of flour. Secondly, as much SS and C flours had to be included as possible and as little W as possible while ensuring the dough’s workability for flatbread production. The experimental design (F1–F9) is reported in Table 1. The three-replicate centre point (Exp. F9–F11) allowed the calculation of experimental error in the analyses and predicted whether the experimental design would give a significant lack of fit [17].

The chosen responses were the texture flatbread quality attributes: force at rupture (f, N), extensibility (e, mm), moisture content (MC, g water/100 g sample), and water activity (*a_w_*). The force at rupture and extensibility were measured by both a puncture test (P) and a one-dimensional extensibility test (E) (TA.XT2 Texture Analyzer, Stable Micro Systems, Godalming, UK). 

#### 2.2.2. Flatbread Production 

Flatbreads were produced as follows: the dry ingredients were mixed at room temperature (R/T) in a home bread-maker (Backmeister 68511, UNHOLD, Hockenheim, Germany) for 2 min; then, sunflower seed oil and salt (2% *w*/*w*) were added and mixed for a further 2 min. Finally, distilled water was added in variable quantities until a dough with an optimal consistency (empirically evaluated), similar to that of the centre point, was obtained. After a resting period of 5 min at R/T, the dough was manually shaped into 50 g balls, which were rested for an additional 25 min at R/T and then laminated twice to obtain a circle of dough (1.85 ± 0.12 mm thickness). The shaped pieces of dough were then cooked at 200 °C on a glass–ceramic skillet (Schott Ceran, Mainz, Germany) for 1 min per side. The flatbread was left to cool at R/T prior to analysis.

#### 2.2.3. Texture 

The textural properties of the flatbreads were measured with a texture analyser (TA.XT2 Texture Analyser) equipped with a 25 kg load cell (Stable Micro Systems, Goldalming, UK) using a puncture test and a one-dimensional extensibility test [18,19]. The rupture force (N) and rupture distance (mm) were recorded for both tests. At least 5 replicates were performed for each test and each flatbread prototype.

#### 2.2.4. Moisture Content (MC) and Water Activity (a_w_) 

The moisture content was measured in triplicate by weight loss after forced-air oven-drying at 105 °C until a constant weight was achieved (ISCO NSV 9035, ISCO, Milan, Italy). The analyses were performed in triplicate. 

Water activity was measured at 25 °C using an Aqualab 4 TE (Decagon Devices, Inc., Pullman, WA, USA). A minimum of 6 determinations were carried out for each flatbread prototype. 

### 2.3. Choice of Formulations: Technological, Nutritional and Economic Criteria 

The experimental design generated weak models characterized by high mathematical uncertainty and low predictivity (see Section 3.1). This prevented us from drawing firm conclusions and from choosing the best formulation from a technological point of view. We, therefore, decided to produce a 100% W-based flatbread taken as a reference (STD) for the evaluation of the textural properties of the composite flour flatbreads. The W flatbread was prepared by mixing 265 g of flour with 187.5 mL of water following the recipe described above (Section 2.2.2).

Consequently, the formulations produced by the experimental design were evaluated based on the textural properties of the derived flatbreads. Moreover, we decided to also consider their estimated nutritional value and costs following the methods described below. The formulations showing the best performance for these three criteria were selected for further characterization.

Physico-chemical properties

The textural properties P_f (force at rupture from puncture test), P_e (extensibility from puncture test), E_f (force at rupture from one-dimensional extensibility test), and E_e (extensibility from one-dimensional extensibility test) of flatbreads were compared with those measured for the STD flatbread. The formulation with textural properties more similar to those recorded for STD was considered eligible to satisfy the technological criterion and therefore selected for further characterizations. 

Nutritional value

An estimated nutritional evaluation of the formulations was carried out. Firstly, the composition of macro and micronutrients expressed in 100 g of flour was calculated based on the nutritional information obtained analytically (Section 2.1, [16]) integrated with information obtained from the FAO/INFOODS Food Composition Table for Western Africa 2019 [20]. Then, for each nutrient, the range of variation identified within the experimental set (F1–F9) was established. For each range and nutrient, 4 target levels were defined: minimum intake, intake equal to 33% of the range, intake equal to 66% of the range, and maximum intake. Lastly, for each nutrient of each formulation, the following scores were assigned: 0 = an intake between the min value and ≤33% of the range of variation; 1 = an intake of >33% but ≤66% of the range; 2 = an intake of >66%. The individual scores were added up to obtain the final score for each formulation.

The most prevalent nutritional deficits among females in sub-Saharan Africa were estimated to be protein–energy malnutrition and iron, zinc, and vitamin A deficiencies [21,22]. Therefore, in a second evaluation, the scores obtained for the components mentioned above were multiplied by two. Eventually, the individual scores were added up and the formulation with the highest total score was considered eligible to satisfy the nutritional criterion and, therefore, selected to further characterization. 

Economic sustainability

The economic sustainability of the formulations was evaluated by calculating their prices ($/kg) in different African countries, Sierra Leone, Tanzania, Burundi, and Togo, based on the costs of the single raw materials and their proportion in the blend. This information was collected on-site by local partner institutions and collaborators of the University of Parma. The least expensive formulation that met the economic sustainability criterion was selected for further analysis.

### 2.4. Characterization of Selected Flatbreads Formulations 

#### 2.4.1. Relevant Starch Nutritional Fractions 

In vitro starch digestibility (rapidly digestible starch (RDS), slowly digestible starch (SDS), and resistant starch (RS)) was quantified with the Megazyme Digestible Starch and Resistant Starch assay procedure (K-DSTRS 11/19 commercial kit, Megazyme International Ireland Ltd., Wicklow, Ireland) following the manufacturer’s protocol based on the method described by Englyst, Vinory, Englyst, and Lang [23]. Two batches of each bread type were produced on different days, and each one was analysed in triplicate. Six determinations of the RDS, SDS, and RS fractions, respectively, were acquired. Digestible starch (DS, RDS + SDS) and total starch (TS, DS + RS) were also calculated.

#### 2.4.2. Total Polyphenol Content (TPC) and DPPH• Free Radical-Scavenging Activity Analysis 

Flatbread phenolic extracts were prepared, starting from 1 g of ground bread added to 20 mL of a methanol/water (70:30 *v*/*v*) mixture, stirred at R/T for 1 h, and then filtered using a paper filter. The solvent was evaporated and the extract was dissolved in 1 mL of a methanol/water (70:30 *v*/*v*) mixture and centrifuged at 5040× *g* (6255 rpm) for 15 min at 4 °C.

For the total phenolic content (TPC) measurement, 50 μL of each extract was mixed with 1160 μL of distilled water, 300 μL of Na_2_CO_3_ aqueous solution (20% *w*/*v*), and 100 μL of Folin–Ciocalteu’s reagent. The absorbance of the solution was measured at 760 nm by a UV-visible spectrophotometer after 30 min of incubation in the dark at R/T. A calibration curve using gallic acid as an external standard (10–300 mg/L) was prepared for quantification. The results were reported in milligrams of gallic acid equivalents (GAE) per kilogram of bread on d.b.

Antioxidant capacity was determined using a DPPH• assay (2,2-diphenyl-1-picrylhydrazyl free radical) following the procedure proposed by Dall’Asta, Cirlini, Morini, Rinaldi, Ganino, and Chiavaro [24]. Briefly, 200 μL of each extract was mixed with 2.6 mL of methanol and 2 mL of a methanolic DPPH• solution (0.2 mmol/L). The absorbance of the solution was measured at 517 nm using a Perkin Elmer UV-Visible spectrophotometer after 30 min of incubation in the dark at R/T. A blank was prepared and analysed following the same procedure. The radical scavenging activity was calculated as follows (Equation (1)):I% = [(Abs_0_ − Abs_1_)/Abs_0_] × 100(1)
where Abs_0_ and Abs_1_ are the absorbances of the blank and the sample, respectively.

The TEAC value (Trolox-equivalent antioxidant capacity; μmol Trolox eq./g of bread on d.b.) was obtained from the 0.1–0.5 mmol/L calibration curve calculated by measuring the absorbance at 517 nm of Trolox ((±)-6-Hydroxy-2,5,7,8-tetramethylchromane-2-carboxylic acid) methanolic solutions at different concentrations. Two batches from each bread type were analysed and the analyses were performed in duplicate.

#### 2.4.3. Colour Analysis 

Colour was measured with a CieLAB colourimeter (CM 2600d, Minolta Co., Japan) using a standard illuminant D65. The *L** (lightness), *a** (degree of redness), and *b** (degree of yellowness) parameters were measured using a 10° position of the standard observer. At least 15 determinations were performed by analysing the surface of three flatbread samples for each prototype. Differences in colour between flatbread prototypes and the STD sample were evaluated using the ΔE value, calculated as follows (Equation (2)):ΔE = [(Δ*L**)^2^ + (Δ*a**)^2^ + (Δ*b**)^2^]^1/2^(2)
where Δ*L** is the difference between *L** values of flatbread prototype and STD; Δ*a** is the difference between *a** values of flatbread prototype and STD; Δ*b** is the difference between *b** values of flatbread prototype and STD.

#### 2.4.4. Preliminary Sensory Evaluation 

Consumers’ sensory evaluation of flatbreads was assessed by means of an acceptability test, a preference ranking test, and a just-about-right (JAR) test. 

The flatbreads were produced a few hours before the analysis. After baking and cooling, discs with a 5 cm diameter (Appendix A) were obtained using a pastry cutter; then, they were divided in half and identified with random three-digit codes. Samples were simultaneously presented on a plate in randomized order and under blind conditions. Consumers were asked to taste and evaluate one sample at a time. Water was served to cleanse the palate between tastings. Prior to starting the sensory test, all interviewees were asked to read and voluntarily sign an informed consent form. A detailed description of the method, duration, and purpose of the study was provided. Moreover, the anonymity of the data and non-use for commercial purposes were guaranteed. Among the interested volunteers, people in a good and healthy state, without allergies nor dietary restrictions, and who did not have any aversions to specific food products were selected as panel members. The panel consisted of 27 untrained African consumers, of which 17 were students at the University of Parma and 10 were recruited among the workers of a local farm. Due to the difficulty in recruiting a larger number of consumers during the COVID-19 emergency period, the number of panellists was limited. Before the analyses, the consumers were asked questions about their age, gender, and country of origin. First, the panellists evaluated the acceptability of the flatbread by rating its overall acceptability, consistency, appearance and aroma by means of a 9-point hedonic scale (1 = dislike extremely, 2 = dislike very much, 3 = dislike, 4 = dislike slightly, 5 = neither like nor dislike, 6 = like slightly, 7 = like, 8 = like very much, and 9 = like extremely). Then, for the JAR test, consumers rated the samples on a 5-point JAR scale (1 = much too low, 2 = too low, 3 = just about right, 4 = too much, and 5 = far too much) for hardness, darkness, bitterness, and bean flavour. After tasting all the samples, the consumers were asked to answer a preference ranking test, before and after providing them with a brief explanation of the project and samples.

### 2.5. Statistical Analysis 

All data were expressed as the mean ± standard deviation (SD). MODDE 12.1 Software (Umetrics AB, Malmö, Sweden) and the PLS method were used to design and analyse the experiments and analyse the results, respectively. The model validity and reproducibility were also evaluated. Reproducibility was evaluated by the variation in the response under the same conditions (pure error), often at the centre points, compared with the total variation in the response. To interpret the influence of terms (factors) on each model, the coefficient plot and effect plots and lists were evaluated. Each model was refined by removing outliers and/or insignificant terms, and/or adding significant/interaction terms. The reliability of the models was then evaluated by calculating the R^2^ and Q^2^ values, where R^2^ is the percentage of the variation in the response explained by the model and Q^2^ is the percentage of the variation in the response predicted by the model according to cross validation, expressed using the same units as R^2^.

One-way analysis of variance (ANOVA) followed by Duncan’s post hoc test at 0.05 significance level (SPSS Statistical Software, Version 25.0, IBM SPSS Inc., Milan, Italy) were performed to assess significant differences between the samples. The Pearson correlation coefficient was also calculated to investigate the statistical correlations (*p* < 0.05) between the antiradical activity and TPC.

A Friedman nonparametric test followed by multiple pairwise comparisons using Nemenyi’s procedure was used to determine the significance of the ranking preferences. For the data analysis of the JAR test, the impact of the JAR variables on the overall liking scores was calculated using Spearman’s correlation coefficient. Then, penalty analysis (PA) was performed grouping the 5-point JAR scale into 3 levels (too little, JAR, and too much) for each attribute. Then, the mean overall liking score from the hedonic scale scores and the percentage of respondents represented in each of the 3 categories were calculated. Lastly, mean drops were determined by subtracting the mean overall liking score for the JAR group from the mean overall liking score of the “too much” or “too little” categories [25]. The penalty was then calculated as the weighted difference between the means (mean of liking for JAR—mean of liking for the two other levels taken together).

The ranking and JAR sensory data analyses were performed using XLSTAT Software (Version 2020.3.1, Addinsoft, New York, NY, USA).

## 3. Results and Discussion 

In a previous study [15], a composite flour made up of SS, T, C, and W at 25% *w*/*w* flour basis (f.b.) was proven to offer a good compromise between technological and nutritional qualities. The aim of this work was to investigate the use of SS_C_T_W composite flour in breadmaking by evaluating the physico-chemical, nutritional, and sensory properties of the finished product while investigating a possible modulation of the ingredients to enhance the sustainability, texture, and nutritional quality of flatbread formulations. 

After preliminary trials to increase the levels of SS, C, and T (locally sourced raw materials) and decrease the level of W (imported flour in Africa), and evaluating the workability of the derived dough, an experimental design was set for screening purposes to investigate the effect of the SS, T, C and W levels on the textural properties of the derived flatbreads.

### 3.1. Experimental Design 

The experimental design allowed homogeneously testing a domain defined by four controlled factors (SS, T, C, W) and one uncontrolled factor (water), i.e., five dimensions. Very low reproducibility was found for the MC and *a_w_* responses; the variability in these data was, overall, comparable with the experimental error determined on the three-replicate centre point. For this reason, these responses were not investigated further.

For all the other answers, the reproducibility of the data was adequate to further study the obtained models; however, they were weak and characterized by high mathematical uncertainty. For all the results, the effects of the factors had to be read in probabilistic terms.

The models generated were generally characterized by low Q^2^ values (Appendix A; additional information about the significance of the models generated can be found in Appendix B). As drawing conclusions from a model with Q^2^ > 0.5 is recommended [17], the low predictivity of the models and the high uncertainty of the effects prevented us from using the model’s mathematical equations to make predictions. For this reason, with all the formulations proposed by the experimental design being workable, three of them were selected for complete characterization. The selection was performed based on three criteria: textural quality, nutritional value (estimated), and economic sustainability.

### 3.2. Choice of Formulations: Technological, Nutritional, and Economic Criteria 

Physico-chemical properties

The textural properties, MC, and *a_w_* of flatbread samples are reported in Table 2. The STD flatbread exhibited the highest puncture rupture force (~8 N) and extensibility (~13 mm), indicating a more cohesive and extensible structure probably due to the presence of a higher amount of gluten proteins in this formulation when compared with flatbreads obtained from composite flours. Sample F6 was the only one that came close to STD with regards to P_f (~6 N) and, together with F4, also in terms of P_e (~13 mm for both F6 and F4). The data obtained from the one-dimensional extensibility test showed, for the STD sample, an intermediate rupture force (~6 N) when compared with the range identified for the other samples (~5 N (F2)–10 N (F5)). On the other hand, the extensibility of the STD (~7 mm) was significantly higher than those of all other formulations. F1 and F6 were characterized by extensibility more similar to that of STD, although significantly (*p* ≤ 0.05) lower (~3 mm for both samples). 

The moisture content of STD was 30.7 g water/100 g sample, which is within the range identified for composite flatbreads (26.1 (F9)—33.3 g water/100 g sample (F8)). Despite F1, F3, F5, F6, and F10 exhibiting similar MC when compared with STD, most of the samples showed significantly (*p* ≤ 0.05) lower *a_w_*, which may be related to stronger interactions between biopolymers and water in these samples, when compared with STD. 

Overall, the changes in the textural attributes of flatbreads may be related to the different water–solids interactions developed as a function of the different compositions (gluten and non-gluten proteins, starch, sugars, fibre, etc.) of the used flours and to their different percentages in the blend. Polymers of different chemical and/or botanical natures can interact with water in different ways [26] and compete for the formation of molecular bonds with it, leading to different levels of polymer hydration. These phenomena significantly affect the water state and distribution in complex matrices, influencing the textural attributes [27]. F6 showed textural attributes more similar to those of STD; thus, it was identified as the sample meeting the technological criterion.

Nutritional value

The estimated proximate composition of the flatbread formulations is presented in Table 3. An initial evaluation of the formulations was provided with scores (0, 1 or 2) based on the theoretical intake provided for each nutrient when compared with the range of variation within the experimental set. Based on this attribution, formulations F5, F6, and F7 exhibited the highest score for the energy and macronutrients (8) and micronutrients (26, 28, and 30, respectively) contents. Overall, F7 was evaluated as the formulation with the best theoretical nutritional profile (total score = 38), followed by F6 (total score = 36) and F5 (total score = 34). 

Previous findings revealed how the most prevalent nutritional deficits among females in sub-Saharan Africa were protein–energy malnutrition and iron, zinc, and vitamin A deficiencies [21,22]. Therefore, a second evaluation was then proposed by attributing a greater relevance to the contribution of the aforementioned components, multiplying the respective scores by two. The results are reported in brackets in Table 3. Formulations F5 and F6 received the highest rating (12) for the contribution of energy and macro components, followed by F7 (10), which differed from the other two mainly due to its lower energy intake. The theoretical highest score for micro components, on the other hand, was attributed to F7 (36), followed by F6 (34) and F5 (32). Overall, F6 and F7 were evaluated as the formulations with the best theoretical nutritional profile (total score = 46), followed by F5 (total score = 44). Their highest theoretical nutritional profile was probably the result of their high content of nutrient-rich flours (e.g., SS and C made up more than 85% of the flour in F7) [13,14,15,16]. Comparing F6 and F7 with STD, the former flours provided a theoretically higher intake of proteins and most micronutrients than the latter, along with lower intake of energy, carbohydrates, fat, fibre, iron, phosphorus, and vitamin B6. As a result, F6 and F7 were selected for further analysis based on the nutritional criterion. Interestingly, F6 was already chosen, given its compliance with the technological criterion. 

Economic sustainability

The prices ($/kg) of the raw flours (W, T, C, and unsprouted sorghum (S)) used for the creation of blends are shown in Table 4. For sorghum, we reported the price of the native flour, because the sprouted flour is not commonly sold at local markets, but it is produced domestically by subjecting sorghum kernels to sprouting and milling. As sprouting represents a feasible and inexpensive technology traditionally performed at the domestic level in Africa [8,28], a similar cost is expected for the raw and sprouted flours. 

For all four considered African countries, T and S were the flours with the lowest price. The price of T varied between a minimum of 0.31 $/kg (Burundi) and a maximum of 1.10 $/kg (Togo); on the other hand, the price of S varied from 0.66 $/kg (Sierra Leone) to 1.29 $/kg (Tanzania). 

In all countries, the cost of C was higher than that of T and S, varying from a minimum of 0.71 $/kg (Sierra Leone) to a maximum of 2.15 $/kg (Tanzania). Although cowpea is a widely cultivated and consumed legume in Africa [14], the higher cost of the derived flour could be associated with the fact that cowpeas are mainly consumed as a legume and not in the form of flour. The prices of W were also higher than those of S and T, as expected, as it is imported. W can be purchased for a minimum price of 0.71 $/kg (Sierra Leone) up to a maximum of 1.38 $/kg in Tanzania. 

The costs of the formulations were a projection of the percentage of each ingredient in the blend and their prices in a specific country. The formulations proposed by the experimental design may have an average price of 0.68 $/kg (Sierra Leone), 1.62 $/kg (Tanzania), 0.67 $/kg (Burundi), and 0.93 $/kg (Togo). F4 was the least expensive composite formulation, thus meeting the economic sustainability criterion. Interestingly, all composite formulations were less expensive than STD (100% W) in Sierra Leone, Burundi, and Togo, while the opposite was observed for Tanzania.

### 3.3. Characterization of Selected Flatbreads 

Based on the previous results, flatbreads F4, F6, and F7 were selected for further analysis and comparison with STD.

#### 3.3.1. Relevant Starch Nutritional Fractions 

Relevant nutritional starch fractions were identified (Table 5) based on the rate of glucose release and its absorption by the gastrointestinal tract during digestion. These include RDS, SDS, and RS, defined as the three sequential nutritional starch fractions determined by the reaction time when in vitro starch digestion is performed [29].

RDS is the starch portion digested in the mouth and in the small intestine, causing a rapid rise in the blood glucose concentration after the consumption of carbohydrates. The RDS fraction determined in vitro identifies the amount of starch digested within 20 min in a standard digestion reaction mixture [30]. The results for RDS were fairly close to each other and all significantly (*p* ≤ 0.05) lower in the blended flour flatbread samples (≈37 g/100 g d.b.) than in STD (≈48 g/100 g d.b.). SDS represents the starch portion digested slowly, but completely, in the human small intestine after RDS and identifies the starch portion digested within more than 120 min under standard in vitro substrate and enzyme concentration conditions [30]. SDS offers potential health benefits, e.g., stable glucose metabolism, control of diabetes, mental performance, and feelings of satiety [31]. The results for SDS in the composite flour-based breads ranged from ≈28 to ≈23 g/100 g d.b.; in all cases, they were higher than that in STD, which exhibited the lowest amount of SDS (≈17 g/100 g d.b.).

Focusing on the percentages of RDS and SDS expressed ion DS, i.e., the starch fraction available for digestion within 120 min [30], F7 exhibited the lowest RDS percentage (and consequently the highest SDS percentage), followed by F6 and F4, which presented intermediate results, and STD, most of whose starch was rapidly digested (73%). 

RS represents the starch portion escaping digestion in the small intestine [30] that may undergo bacterial fermentation in the colon. The composite flatbreads contained RS amounts ranging from ≈0.93 (F4) to ≈0.69 (F6) g/100 g d.b., which resulted, in all cases, significantly higher values than the RS amount measured in the STD sample (0.48 g/100 g d.b.).

Overall, all composite flatbreads exhibited lower RDS and higher RS values than STD. These traits are of relevance from a nutritional point of view [23]. As T is a source of readily available carbohydrates [11], and sprouting has been proven to increase starch digestibility [16,32,33], the results observed for blended flour flatbread may be principally attributed to C. The incorporation of wholemeal legume flour into bread recipes appears to decrease starch hydrolysis, possibly due to the lower starch content, but greater fibre and protein contents. In fact, the reduced rates, as well as the overall reduced starch digestibility of legumes, are influenced by the cell wall structure, phenolic compounds, high amylose content, and viscous soluble dietary fibre components. Additionally, the high protein content of legumes can promote starch–protein interactions, which limits enzymatic attack [33,34].

#### 3.3.2. Total Phenolic Content (TPC) and DPPH• Free Radical Scavenging Activity 

The TPC of the flatbread samples is reported in Table 5. F7 showed the highest TPC (≈835 mg GAE/kg d. b), followed by F4 (≈771 mg GAE/kg d. b) and F6 (≈655 mg GAE/kg d. b), the latter not being significantly different (*p* ≤ 0.05) from STD (≈651 mg GAE/kg d. b). The highest TPC measured for flatbread F7 may be related to the highest presence of SS and C in the composite flour used, which possibly enriched the finished product with bioactive compounds with potential beneficial effects on human health. Indeed, as is known from the literature, like other legume flours, cowpea flour is a good source of phenolic compounds [14,35]. In addition, the positive function of sorghum sprouting on the phenolic content of wholemeal flour has been documented in detail in the literature [36,37,38,39,40]. It has been scientifically proven that the species of reactive oxygen and the free radicals generated in cellular metabolism or the peroxidation of lipids and other biological molecules play important roles in chronic diseases, such as coronary heart disease and cancer. Antioxidants in the diet fight against free radicals and probably help to prevent chronic disease risk factors [40]. In this work, the antioxidant capacity of the flatbreads was determined based on the DPPH• free radical scavenging activity and expressed as Trolox-equivalent antioxidant capacity (TEAC); the results are reported in Table 5. The TEAC values were 1.09, 1.03, 1.96, and 0.92 µmol Trolox eq./g d.b. for F7, F4, F6, and STD, respectively, the latter two not significantly differing from each other (*p* ≤ 0.05).

As expected, the Pearson correlation revealed a strong positive correlation (r = 0.90; *p* < 0.05) between TPC and TEAC. However, the improved TEAC observed in the composite flour bread samples could be due not only to the enhancement of phenolic compounds in the flatbread, but also to the documented increase in antioxidant vitamins after sorghum sprouting, along with the antioxidant potential of cowpea proteins. These factors are mainly related to the hydrophobic and aromatic amino acids’ capacity to donate protons to free radicals [14] and to potential Maillard reaction products formed during baking [41].

#### 3.3.3. Colour Analysis 

The colour parameters (*L**, *a**, and *b**) of the selected flatbreads and of the STD sample are presented in Table 5. STD exhibited the highest lightness (*L** ≈ 67) and the lowest redness (*a** ≈ 4) compared with the composite flour-based flatbreads, while its yellow hue (*b** ≈ 11) was low, but not significantly different from that of F4 (*b** ≈ 10.6). F4, in turn, had the lowest brightness (*L** ≈ 53), while its red colour component was not significantly different from those of the other flatbreads (*a** ≈ 6.5), which may have been due to the strong reddish-brown notes of the SS and the lighter reddish notes of the C. Among the flatbreads, F6 had lighter coloured notes (*L** ≈ 57), probably due to the lower presence of SS in the blend, and more intense yellow notes (*b** ≈ 14.6), while F7 had intermediate parameters (*L** ≈ 54.6 and *b** ≈ 13.1). Overall, the lower brightness and more accentuated yellow–red hues of the blended flour flatbreads compared with STD may be attributed to the SS and C flours, and possibly to the more extensive Maillard reaction occurring during cooking. 

The ΔE values could estimate whether the colour differences between a sample made from blended flour and STD were perceivable to the human eye (the higher the value, the higher the perceived differences between the samples) [42]. A ΔE value of 11.2 was found for sample F6, leading to a strong colour difference from that of STD, while ΔE values > 12 revealed that samples F4 and F7 were perceived as having a different colour from the STD (Appendix A).

#### 3.3.4. Preliminary Sensory Evaluation 

The consumer panel was composed of 15 men (56%) and 12 women aged between 18 and 45 years (37% aged ≤ 25 years, 44% aged between 26 and 35 years, and 19% aged ≥ 36 years) and coming from Ghana (6), Cameroon (5), Morocco (5), Togo (3), Gabon (3), Tanzania (1), Rwanda (1), Tunisia (1), Gambia (1), and Congo (1). As stated in the Material and Methods section, the number of judges was limited due to difficulties in consumer recruitment during the lockdown period related to the COVID-19 emergency. Nevertheless, given the great relevance of the assessment of sensory properties of these flatbreads among people who regularly consume this kind of product, some preliminary indications may be taken as a basis for further studies. 

The average scores obtained from the blind test of flatbreads are presented in Table 5. Based on a one-way ANOVA, the flatbread formulation significantly affected (*p* ≤ 0.05) the acceptability of all measured sensory attributes (overall acceptability, consistency, appearance, and flavour). The scores recorded in the acceptability test for all attributes revealed that the different kinds of flatbread were generally appreciated by the assessors (scores > 5). The consistency of samples STD, F6, and F7 was evaluated in a similar way by the consumers (average score = 6.3), while the consistency of F4 obtained a significantly lower score (5.2). These results were in agreement with the textural properties measured analytically, especially concerning the similar textural perception recorded for F6 and STD, and the worse textural properties measured for F4 compared with STD. With regard to appearance, STD was the most appreciated (7), followed by F6 (6.5) and F7 (6.4); additionally, for this attribute, F4 exhibited the lowest score (5.2). The flavour results showed the highest scores for both the STD and F7 samples (6.9 and 6.8, respectively), followed by F4 (6.2) and F6 (5.4). The overall acceptability results revealed that STD was the most appreciated sample (7.2), followed by F7 (6.5) and by both F4 and F6 (5.7 and 6, respectively), mainly due to their texture and appearance (F4) or flavour (F6).

Penalty analysis (PA) was performed to define potential pathways to product improvement. After aggregating the JAR scores obtained for the attributes of each sample (Appendix A) into a three-level scale (Appendix A), PA was performed to underscore how many points had been lost due to a product dimension judged “too much” or “too little” by consumers [25]. Appendix A shows the representative PA result (F4 bread). The *p*-value (significance level = 0.05) calculated for all attributes was not significant, indicating that the attributes evaluated at a non-JAR level were not identified as a potential target for product optimization.

The preference ranking data, collected before providing the panellists a description of the flatbreads’ main features, did not reveal any significant differences (*p* ≤ 0.05) between the samples. After an explanation of the project and samples, the Friedman nonparametric test showed significant differences among the preference ranking data (*p* ≤ 0.05). The multiple pairwise comparisons (Nemenyi’s procedure) revealed that the sample meeting the nutritional criteria (F7) was the preferred flatbread, followed by F6, F4. and STD (Appendix A), which was the least preferred, despite the high scores obtained in the acceptability test, possibly due to its perceived lower sustainability. 

## 4. Conclusions

This work aimed to develop high-quality flatbread formulations for low-income countries by using climate-resilient crops. Wheat, tapioca, sprouted sorghum, and cowpea flour blends were used for this purpose. Among the flatbread prototypes proposed, three of them were chosen on the basis of textural, nutritional (based on the estimated composition of macro and micronutrients), and economic criteria. Then, the three selected prototypes were further characterized for their in vitro starch digestibility, total phenolic content, total antioxidant capacity, physico-chemical, and sensory properties, in comparison with a reference 100% wholewheat flatbread (STD). Overall, if the brief is to produce a flatbread with a physico-chemical profile similar to that of the 100% W reference, the use of a composite made of SS, T, C, and W at percentages of 29.2, 12.5, 49.7 and 8.6 *w*/*w* f. b. (F6), respectively, should be preferred. Moreover, this formulation was less expensive than 100% W in Sierra Leone, Burundi, and Togo, and showed delayed starch hydrolysis and similar TPC and antioxidant activity compared with STD.

If the objective is to meet the nutritional criterion, a flour blend composed of SS, T, C, and W at proportions of 42.1, 7.4, 43.1, and 7.4% *w*/*w* f.b. (F7), respectively, should be chosen. Other than having delayed starch digestibility than the 100% W reference, it seemed to provide a higher content of phenolic compounds and antioxidant activity compared with the STD flatbread. Although the derived flatbread textural properties were found to be different from those of the STD when measured analytically, it appears that consumers did not detect significant differences in texture compared with the control, and even the flavour of this flatbread was perceived as acceptable as that of STD. However, consumers’ sensory perceptions of flatbreads should be confirmed with a larger panel.

Finally, if the primary objective is economic sustainability, a composite made of SS, T, C, and W at percentages of 48.6, 12.5, 30.3, and 8.6% *w*/*w* f.b. (F4), respectively, should be preferred to others. Additionally, this formulation was proven to have a delayed starch digestibility, a higher content of phenolic compounds and antioxidant activity than STD, but worse textural properties.

Overall, the use of composite flour made from climate-resilient crops in breadmaking was proven to be an efficient strategy to promote the use of locally available raw materials and obtain economically sustainable, nutritionally enhanced flatbread with an acceptable technological and sensory profile.

## Figures and Tables

**Table 1 foods-12-01638-t001:** Experimental design for flatbread development. The percentages of flour expressed in 100 g of composite flour are in brackets.

-	-	X_1_	X_2_	X_3_	X_4_	X_5_
Exp. Name	Run Order	W (g)	T (g)	SS (g)	C (g)	Water (mL)
F1	6	22 (11.2)	22 (11.2)	75 (38.1)	78 (39.5)	112.7
F2	5	32 (14.7)	32 (14.7)	75 (34.6)	78 (36)	110.8
F3	10	32 (12.5)	22 (8.6)	125 (48.6)	78 (30.3)	146.1
F4	8	22 (8.6)	32 (12.5)	125 (48.6)	78 (30.3)	147.8
F5	7	32 (12.5)	22 (8.6)	75 (29.2)	128 (49.7)	142.6
F6	11	22 (8.6)	32 (12.5)	75 (29.2)	128 (49.7)	130.8
F7	4	22 (7.4)	22 (7.4)	125 (42.1)	128 (43.1)	161
F8	9	32 (10.1)	32 (10.1)	125 (39.4)	128 (40.4)	183.3
F9	1	27 (10.5)	27 (10.5)	100 (38.9)	103 (40.1)	140
F10	2	27 (10.5)	27 (10.5)	100 (38.9)	103 (40.1)	140
F11	3	27 (10.5)	27 (10.5)	100 (38.9)	103 (40.1)	140

Levels of wholewheat flour (W) and tapioca (T) = 22, 27, and 32 g; levels of wholemeal sorghum flour (SS) = 75, 100, and 125 g; levels of cowpea wholemeal (C) = 78, 103, and 128 g.

**Table 2 foods-12-01638-t002:** Experimental design results of composite flour formulations (F1-F11) for flatbread development and comparison with the textural properties of a 100% W flatbread (STD).

Trial	Puncture Test		One-Dimensional Extensibility Test		MC (g/100 g)	*a* * _w_ *
-	P_f (N)	P_e (mm)	E_f (N)	E_e (mm)	-	-
F1	3.84 ± 0.1 e	11.45 ± 0.47 abc	8.56 ± 0.39 b	3.26 ± 0.53 b	29.77 ± 0.34 bcde	0.9087 ± 0.0013 bcd
F2	2.38 ± 0.28 f	10.26 ± 0.42 cd	5.47 ± 0.8 f	3.06 ± 0.46 bc	29.34 ± 0.01 de	0.9104 ± 0.0009 bc
F3	4.78 ± 0.95 cde	11.31 ± 1.95 abc	7.54 ± 1.32 bc	2.76 ± 0.6 bc	30.93 ± 0.31 b	0.9064 ± 0.0008 cde
F4	2.92 ± 0.7 4 f	12.81 ± 1.32 a	5.98 ± 1.32 ef	2.11 ± 0.63 d	32.74 ± 0.68 a	0.9184 ± 0.0041 a
F5	5.37 ± 0.63 bc	10.72 ± 1.05 bcd	10.01 ± 0.73 a	3.1 ± 0.51 bc	30.03 ± 0.45 bcd	0.9028 ± 0.0042 de
F6	5.89 ± 1.13 b	12.64 ± 0.37 ab	7.43 ± 1.12 bcd	3.23 ± 0.54 b	29.57 ± 0.52 cde	0.8964 ± 0.002 f
F7	2.51 ± 0.52 f	9.76 ± 1.34 cd	6.88 ± 1.57 cde	2.6 ± 0.38 c	32.45 ± 0.47 a	0.9173 ± 0.0003 a
F8	3.99 ± 0.41 e	11.41 ± 1.67 abc	7.42 ± 1.6 bcd	2.75 ± 0.54 bc	33.3 ± 0.02 a	0.9148 ± 0.0031 ab
F9	4.99 ± 1.07 bcd	9.95 ± 0.35 cd	6.43 ± 1.22 cdef	2.73 ± 0.5 bc	26.1 ± 0 f	0.884 ± 0.009 g
F10	4.51 ± 0.84 cde	9.29 ± 0.85 d	6.25 ± 0.6 def	2.64 ± 0.28 c	29.52 ± 0.89 cde	0.9021 ± 0.0036 ef
F11	4.36 ± 0.38 de	10.05 ± 0.73 cd	7.12 ± 0.82 cde	3.1 ± 0.31 bc	28.66 ± 1.36 e	0.9015 ± 0.0027 ef
STD	7.78 ± 1.26 a	12.91 ± 3.32 a	6.12 ± 1.02 ef	7.32 ± 0.6 a	30.7 ± 0.28 bc	0.9132 ± 0.0010 ab

Values are expressed as the mean ± SD (*n* ≥ 5; *n* = 3 for moisture content, MC). Values followed by different letters in each column are significantly different (one-way ANOVA with Duncan’s post hoc test. *p* ≤ 0.05). P_f (force at rupture from puncture test); P_e (extensibility from puncture test); E_f (force at rupture from one-dimensional extensibility test); E_e (extensibility from one-dimensional extensibility test).

**Table 3 foods-12-01638-t003:** Estimated macro and micronutrient composition of flatbread formulations (F1-F9) and of a 100% W flatbread (STD), per 100 g of flour. In brackets, the score attributed to the theoretical intake of each nutrient provided by the formulation is presented. The total score is the sum of the partial scores.

Component (100 g)	F1	F2	F3	F4	F5	F6	F7	F8	F9	STD
Energy (kcal)	206 (2)	219 (4)	172 (0)	171 (0)	235 (4)	235 (4)	192 (0)	201 (2)	203 (2)	354
Protein (g)	14.45 (2)	13.74 (0)	13.58 (0)	13.17 (0)	15.94 (4)	15.54 (4)	15.15 (4)	14.62 (2)	14.56 (2)	12.00
Fat (g)	1.94 (1)	1.89 (0)	2.09 (2)	2.03 (2)	1.86 (0)	1.80 (0)	1.99 (2)	1.96 (1)	1.95 (1)	2.2
Carbohydrate (g)	63.42 (1)	64.27 (2)	65.37 (2)	65.94 (2)	60.69 (0)	61.26 (0)	62.61 (0)	63.24 (1)	63.31 (1)	65.3
Fibre (g)	9.65 (1)	9.52 (0)	9.31 (0)	9.00 (0)	10.32 (2)	10.01 (2)	9.77 (2)	9.67 (1)	9.66 (1)	12.2
Ash (g)	2.12 (1)	2.07 (0)	1.90 (0)	1.92 (0)	2.34 (2)	2.35 (2)	2.18 (2)	2.13 (1)	2.13 (1)	1.3
Ca (mg)	46.59 (1)	47.41 (2)	38.35 (0)	39.16 (0)	53.42 (2)	54.23 (2)	45.47 (0)	46.10 (1)	46.29 (1)	45.00
Fe (mg)	3.40 (2)	3.40 (2)	2.89 (0)	2.78 (0)	4.01 (4)	3.89 (4)	3.41 (4)	3.38 (0)	3.39 (2)	4.9
Mg (mg)	145.64 (1)	137.70 (0)	131.53 (0)	130.87 (0)	162.73 (2)	162.06 (2)	153.45 (2)	147.52 (1)	146.80 (1)	68
P (mg)	203.57 (1)	200.85 (0)	165.54 (0)	160.09 (0)	246.47 (2)	241.03 (2)	205.06 (2)	203.10 (1)	203.28 (1)	244
K (mg)	604.43 (1)	574.76 (0)	527.19 (0)	521.47 (0)	695.07 (2)	689.35 (2)	632.75 (2)	610.65 (1)	608.27 (1)	356
Na (mg)	9.80 (1)	9.27 (0)	7.81 (0)	7.73 (0)	11.99 (2)	11.91 (2)	10.29 (2)	9.89 (1)	9.86 (1)	5.00
Zn (mg)	2.60 (2)	2.46 (0)	2.56 (0)	2.49 (0)	2.76 (4)	2.69 (4)	2.75 (4)	2.64 (2)	2.63 (2)	2.00
Cu (mg)	0.52 (1)	0.49 (0)	0.51 (0)	0.50 (0)	0.56 (2)	0.55 (2)	0.56 (2)	0.53 (1)	0.53 (1)	0.27
Vit A (µg RE)	0.79 (2)	0.72 (0)	0.61 (0)	0.61 (0)	1.00 (4)	1.00 (4)	0.86 (4)	0.81 (2)	0.80 (2)	0
Vit E (mg)	1.07 (1)	0.99 (0)	1.22 (2)	1.21 (2)	0.97 (0)	0.96 (0)	1.17 (2)	1.10 (1)	1.09 (1)	0.23
Vit C (mg)	0.78 (1)	0.76 (0)	0.60 (0)	0.64 (0)	0.93 (2)	0.97 (2)	0.81 (2)	0.79 (1)	0.79 (1)	0
Thiamine (mg)	0.39 (1)	0.38 (0)	0.37 (0)	0.36 (0)	0.43 (2)	0.42 (2)	0.41 (2)	0.39 (1)	0.39 (1)	0.37
Riboflavin (mg)	0.12 (1)	0.12 (0)	0.12 (0)	0.12 (0)	0.13 (2)	0.13 (2)	0.13 (2)	0.12 (1)	0.12 (1)	0.09
Niacin (mg)	3.36 (1)	3.27 (0)	3.75 (2)	3.68 (2)	3.09 (0)	3.02 (0)	3.47 (2)	3.40 (1)	3.38 (1)	3.3
Pantothenic acid	1.05 (1)	0.95 (0)	1.34 (2)	1.34 (2)	0.81 (0)	0.81 (0)	1.16 (2)	1.09 (1)	1.07 (1)	n.a.
Vitamin B6 (mg)	0.41 (2)	0.43 (2)	0.40 (0)	0.41 (1)	0.40 (0)	0.41 (2)	0.39 (0)	0.40 (1)	0.41 (1)	0.49
Folate (µg)	191.83 (1)	180.60 (0)	153.66 (0)	155.99 (0)	230.31 (2)	232.65 (2)	202.32 (2)	193.97 (1)	193.15 (1)	40
Total score	29	12	10	11	44	46	46	26	28	-

RE, retinol equivalent; n.a., not available.

**Table 4 foods-12-01638-t004:** Price ($/kg) of flatbread formulations based on raw material (W, T, S, and C flours) costs in Sierra Leone, Tanzania, Burundi, and Togo. W, wholewheat flour; T, tapioca; S, unsprouted sorghum flour; C, cowpea wholemeal flour.

Flour	Price ($/kg)			
-	Sierra Leone	Tanzania	Burundi	Togo
W	0.71	1.38	0.79	1.17
T	0.59	1.08	0.31	1.10
S	0.66	1.29	0.68	0.50
C	0.71	2.15	0.73	1.26
F1	0.68	1.62	0.67	0.94
F2	0.68	1.58	0.66	0.96
F3	0.68	1.54	0.68	0.87
F4	0.67	1.53	0.66	0.86
F5	0.69	1.71	0.69	1.01
F6	0.68	1.70	0.67	1.01
F7	0.68	1.65	0.68	0.92
F8	0.68	1.63	0.67	0.94
F9	0.68	1.62	0.67	0.94

**Table 5 foods-12-01638-t005:** Colour, sensory, and nutritional properties of flatbreads. In brackets, the percentage of RDS or SDS calculated in the DS is presented.

-	F4	F6	F7	STD
Colour ^a^				
*L**	53.3 ± 1.71 d	57.11 ± 1.97 b	54.63 ± 1.26 c	67.37 ± 1.87 a
*a**	6.38 ± 0.62 a	6.58 ± 0.93 a	6.89 ± 0.68 a	3.8 ± 0.54 b
*b**	10.59 ± 1.24 c	14.62 ± 1.71 a	13.15 ± 1.17 b	10.94 ± 0.84 c
ΔE (STD as the reference)	14.3	11.2	13.3	-
**Sensory analysis ^d^**				
**Acceptability test**				
Consistency	5.22 ± 0.97 b	6.26 ± 1.06 a	6.15 ± 0.91 a	6.33 ± 0.92 a
Appearance	5.89 ± 1.01 c	6.48 ± 1.09 ab	6.44 ± 0.89 b	7 ± 0.83 a
Flavour	6.15 ± 1.03 b	5.44 ± 0.97 c	6.78 ± 0.97 a	6.89 ± 0.89 a
Overall acceptability	5.7 ± 0.99 c	5.96 ± 1.09 c	6.48 ± 0.94 b	7.19 ± 0.68 a
**Nutritional fractions of starch (g/100 g d.b.) ^b^**				
RDS	35.51 ± 2.26 b (61)	37.56 ± 1.68 b (62)	36.96 ± 0.79 b (57)	47.55 ± 3.43 a (73)
SDS	23.07 ± 1.11 ab (39)	22.82 ± 2.66 ab (38)	27.87 ± 1.41 a (43)	17.3 ± 5.62 b (27)
DS	58.58	60.38	64.83	64.85
RS	0.93 ± 0.01 a	0.69 ± 0.04 c	0.79 ± 0.08 b	0.48 ± 0.02 d
TS	60	61	66	65
**TPC (mg GAE/kg d.b.) ^c^**	770.78 ± 0.69 b	654.78 ± 1.25 c	835.45 ± 1.40 a	650.65 ± 1.20 c
**TEAC (** **µmol Trolox eq./g d.b.) ^c^**	1.03 ± 0.02 b	0.96 ± 0.03 c	1.09 ± 0.03 a	0.92 ± 0.03 c

^a^ *n* ≥ 10; ^b^ *n* = 6; ^c^ *n* = 4; ^d^ *n* = 27. Values followed by different letters in each row are significantly different (one-way ANOVA with Duncan’s post hoc test. *p* ≤ 0.05). F4, formulation chosen on the basis of the economic sustainability criteria; F6, formulation chosen on the basis of the textural criteria; F7, formulation chosen on the basis of the nutritional criteria; JAR, just-about-right test; RDS, rapidly digestible starch; SDS, slowly digestible starch; DS, digestible starch (RDS + SDS); RS, resistant starch; TS, total starch (DS + RS); TPC, total polyphenol content; GAE, gallic acid equivalent; TEAC, Trolox-equivalent antioxidant capacity.

## Data Availability

The data presented in this study are available on request from the corresponding author.

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
