# Peer review of "Towards Sustainable and Nutritionally Enhanced Flatbreads from Sprouted Sorghum, Tapioca, and Cowpea Climate-Resilient Crops"

_foods, 2023, doi:10.3390/foods12081638_

Round 1
Reviewer 1 Report
After carefully reading the manuscript entitled: "Towards sustainable and nutritionally enhanced flatbreads from sprouted sorghum, tapioca and cowpea climate-resilient crops" it can be concluded that the authors spent a lot of time and effort in conducting experiments and writing an article. The topic is interesting. However, some things need to be worked through and explained. Below are remarks, clarification requests, and suggestions.
1. Extensive editing of the English language, grammatic, and style is required.
2. Some sentences are too long, and the terminology used is not adequate.
3. Abstract missing important information, reconsider to extend in some
4. References are not in good order 1, 2, and then 5 (3 and 4 appearing later in the text)
5. Introduction essential parts are missing: background of sustainability in aiming countries, economical part regarding flours in target countries
6. The proximate composition of SS flour was…
7. Tapioca (T) and cowpea wholemeal (C) flours were purchased from Molino Bon Giovanni S.r.l. (Villanova Mondovì, CN, Italy), while wholewheat flour… Why have all flours been purchased? There is no data about those flours' manufacturing or technical characteristics. At some moment, it was assumed that some flour was fortified with some substances. Chemical analysis is missing in order to have any conclusion; otherwise it is just assumptions.
8. Table 1. Suma for F1 of used flour is 100,2 %
9. Nutritional value
-A nutritional evaluation of the formulations was carried out. Firstly, the composition in macro and micronutrients expressed on 100 g of flour was calculated based on the nutritional information obtained analytically on SS – there was no analytical investigation. Data was used from another paper.
-integrated with information obtained from the FAO / INFOODS Food Composition Table for Western Africa 2019 – again assumptions.
10. Conclusions are not based on the evidence. Many statements are made on assumptions.
Based on the above, the paper needs significant revision.
Author Response
We would like to thank the reviewer for her/his comments and suggestions. We have revised the manuscript accordingly.
After carefully reading the manuscript entitled: "Towards sustainable and nutritionally enhanced flatbreads from sprouted sorghum, tapioca and cowpea climate-resilient crops" it can be concluded that the authors spent a lot of time and effort in conducting experiments and writing an article. The topic is interesting. However, some things need to be worked through and explained.
Below are remarks, clarification requests, and suggestions.
- Extensive editing of the English language, grammatic, and style is required.
The manuscript has been carefully revised according to the reviewer comment.
- Some sentences are too long, and the terminology used is not adequate.
Also for this comment, authors reviewed the manuscript accordingly.
- Abstract missing important information, reconsider to extend in some
The abstract has been improved.
- References are not in good order 1, 2, and then 5 (3 and 4 appearing later in the text)
The order of the references has been corrected.
- Introduction essential parts are missing: background of sustainability in aiming countries, economical part regarding flours in target countries.
We included in the introduction section additional information about the concept of sustainability related to the use of local crops in food production (Line 65-90_Revised version).
Nevertheless, we would prefer to not deepen the issue related to the economical part regarding flours in target countries. In our opinion, doing so would mean give too much emphasis on this aspect with the risk that the reader would expect more results and discussion about the economical part. We would like to point out that the economic aspects were only one of the criteria used to screen among flatbread formulations. Therefore, we do not intend to shape our manuscript mainly on this topic. Moreover, as it has been highlighted in Table 4, the prices of different flatbread formulations may vary a lot among different African countries. Thus, there are several factors country-specific that influence the economical parts of the flours and this discussion goes beyond the main aim of our study. We will take into consideration this comment for future research and consider investigating the different factors shaping the economy of the flour industry.
- The proximate composition of SS flour was…
Correction has been made.
- Tapioca (T) and cowpea wholemeal (C) flours were purchased from Molino Bon Giovanni S.r.l. (Villanova Mondovì, CN, Italy), while wholewheat flour… Why have all flours been purchased? There is no data about those flours' manufacturing or technical characteristics. At some moment, it was assumed that some flour was fortified with some substances. Chemical analysis is missing in order to have any conclusion; otherwise it is just assumptions.
The flours used in this study had already been characterized for their chemical and technological functionality in a previous paper (Marchini et al., 2021). This second study was developed on the base of the previous results. As requested, the chemical composition of the flours is now reported in the text (Line 121-131_Revised version), while for the technical properties there is a reference to the previous paper.
- Table 1. Suma for F1 of used flour is 100,2 %
Correction has been made.
- Nutritional value
A nutritional evaluation of the formulations was carried out. Firstly, the composition in macro and micronutrients expressed on 100 g of flour was calculated based on the nutritional information obtained analytically on SS – there was no analytical investigation. Data was used from another paper.
The analytical investigation was carried out in the preliminary steps of a wider research project. Indeed, in line 212 this specification is referred to Paragraph 2.1 where it is specified that these data have been already published (Reference [16]). Anyway, to avoid any misunderstanding for the reader, the reference was also added to line 212.
-integrated with information obtained from the FAO / INFOODS Food Composition Table for Western Africa 2019 – again assumptions.
The experimental design used generated weak models, thus preventing us from choosing the best formulation from a technological point of view. Consequently, to meet the objective of this work, i.e., to develop nutritionally enhanced, economically sustainable and quality acceptable flatbreads, the formulations originated by the experimental design were evaluated based on the textural properties of the derived flatbreads and on their estimated nutritional value and costs. The formulations showing the best performance for these three criteria were selected for further characterization. We are aware that the macro and micronutrients composition for all formulations was estimated and not analytically determined. Nevertheless, we believe that this evaluation can be considered proper for the purpose it had, that is the screening among different formulations. Then, “best” formulations were further analytically characterized, also from a nutritional point of view.
To better clarify the different approach used in (i) the first evaluation of formulations and in (ii) the assessment of nutritional traits of selected formulations, further emphasis has been given to the estimated and therefore theoretical nutritional composition of flatbreads formulations carried out in the first evaluation (line 417-445).
- Conclusions are not based on the evidence. Many statements are made on assumptions.
Main conclusions were formulated based on analytically determinations carried out on flatbread prototypes (physico-chemical properties, preliminary sensory assessment, starch digestibility, TPC and TAC).
Where the nutritional criteria used to select flatbread formulations was mentioned, further clarification about the approach used (estimation) to determine the composition in macro and micronutrients was added (Line 624-625).
Based on the above, the paper needs significant revision.
Response to the reviewer is here attached

Reviewer 2 Report
The manuscript "Towards sustainable and nutritionally enhanced flatbreads 2 from sprouted sorghum, tapioca and cowpea climate-resilient 3 crops" is novely and hot topic in food technology, especialy in bread production technology.
I have some sugestion about authors:
Add full name for RH when it first appears
If investigation is approving for ethical committee for food researches, authors need to add number of the approval of ethical committee for food.
In part 3.3.1. Relevant starch nutritional fractions, the authors need to write that the results is presented in table 5.
Author Response
We would like to thank the reviewer for her/his comments and suggestions. We have revised the manuscript accordingly.
The manuscript "Towards sustainable and nutritionally enhanced flatbreads 2 from sprouted sorghum, tapioca and cowpea climate-resilient 3 crops" is novely and hot topic in food technology, especialy in bread production technology.
I have some sugestion about authors:
Add full name for RH when it first appears
Done
If investigation is approving for ethical committee for food researches, authors need to add number of the approval of ethical committee for food.
We thank the reviewer for pointing it out. At the time of the data collection, our institution did not have any Ethical Board Committee for Research in the food science area. At that time, only research in the field of clinical or medical topics were required to ask permission from the Ethical Board. Therefore, we do not have any number of the approval. However, when recruiting participants, we followed the “Declaration of Helsinki – Ethical Principles for Medical Research Involving Human Subjects”, and therefore we provided all the information about the study (risks and benefits) and asked for their consent by signing a form. This has been reported in the text.
In part 3.3.1. Relevant starch nutritional fractions, the authors need to write that the results is presented in table 5.
Done
Response to the reviewer is here attached

Reviewer 3 Report
Comments to the Author
The present study aimed to developing nutritionally enhanced, quality acceptable and economical sustainable flatbreads for low-income countries by using climate resilient crops. I think this is a very meaningful research work. But there are still some issues that need the author to make appropriate revisions and clarifications.
1. Abstract needs careful revision, and it should display some of the most important research findings in the abstract section.
2. The author's purpose in adding polyphenol extracts to bread is what I am curious about. Is the sole purpose of adding these extracts to enhance the bread's antioxidant activity? Is antioxidant activity crucial for the nutrition of bread? Would this increase the cost of bread, making it incompatible with the author's goal of producing economical bread?
3. Authors should cite appropriate references in the methodology section
4. The author should delve deeper into some important characteristics of bread instead of irrelevant and unimportant indicators such as antioxidant activity.
5. The results in the table should be shown as mean±sd
Author Response
We would like to thank the reviewer for her/his comments and suggestions. We have revised the manuscript accordingly.
Comments to the Author
The present study aimed to developing nutritionally enhanced, quality acceptable and economical sustainable flatbreads for low-income countries by using climate resilient crops. I think this is a very meaningful research work. But there are still some issues that need the author to make appropriate revisions and clarifications.
Thank you very much for your positive feedback.
- Abstract needs careful revision, and it should display some of the most important research findings in the abstract section.
Abstract was revised accordingly.
- The author's purpose in adding polyphenol extracts to bread is what I am curious about. Is the sole purpose of adding these extracts to enhance the bread's antioxidant activity? Is antioxidant activity crucial for the nutrition of bread? Would this increase the cost of bread, making it incompatible with the author's goal of producing economical bread?
We believe that the use of composite flours in foods development in low-income countries such as African ones can be a valuable strategy to (1) save money by reducing importation of wheat flour, (2) provide a better supply of macro and micronutrients in the diet, (3) ameliorate the use of local agriculture productions by employing indigenous climate resilient crops offering many more business opportunities along the value chain. Considering the advantage (2), we tried to formulate flatbreads with higher nutritional value than wheat-based control. Our purpose was to do it using African local crops such as sorghum, tapioca, and cowpea. Sorghum is a key dryland food crop cultivated in marginal lands in more than 100 countries and over 60% of global sorghum production comes from developing countries (Africa and Asia). Sorghum composition includes dietary fibre, fat-soluble and B-vitamins, minerals, and polyphenols. Sorghum sprouting is an empirically and widely used treatment carried out at household level in low-income countries. This treatment may further increase in sorghum bioactive components, such as vitamins, minerals and phenolic compounds, and also increase their availability by reducing antinutritional factors. Cowpea is a cheap source of protein, amino acid lysine, carbohydrate, fibre and bioactive compounds.
In our work, we formulated flatbreads with “theoretically” (because of an estimated composition) improved nutritionally value considering both macro and micronutrients if compared with conventional wheat-based product. Additionally, we analytically investigated some nutritional traits such as starch fractions, polyphenol contents and antioxidant activity. We agree with the reviewer about the limited information belonging to the antioxidant activity measurement, surely not crucial! Anyway, it may give us an indirect information about the presence of high amount of molecules which may have a beneficial effect on human health. Moreover, we have to consider the populations we are “moving” on, that is where nutritional issues e.g., deficits about protein-energy, Iron, Zinc and Vitamin A, occur. All the nutrients and compounds that can be increased in staple food such as flatbread may have a significant impact on the diet of these people.
About the economic issue, it was estimated lower costs (in specific countries considered in this study) for flatbreads formulated with local crops than the one with the imported wheat.
- Authors should cite appropriate references in the methodology section
Which are the not appropriate references we used in the methodology section?
We apologize about the question but in the methodology section we have used:
Line 116 à Ref [16]: it is needed to inform the reader about the ratio used to select sorghum sprouting conditions and to inform the reader about the sorghum proximate composition data already published.
Line 148 à Ref [17]: it was used to support the choice to perform three-replicate central points.
Line 175 à Ref [18,19]: they were used to support the choice of the texture methods.
Line 214 à Ref [20]: it is needed to show the database used to estimate the composition of cowpea, tapioca flours.
Line 223 à Ref [21,22]: they were used to support the sentence “the most prevalent nutritional deficits among females in sub-Saharan Africa…”.
Line 242 à [23]: corresponding to the method of Englyst
Line 263 à [24]: it was used to support the method used for the antioxidant capacity determination
Line 342 à [25]: it was used to support the method to elaborate JAR test.
We kindly ask to the reviewer which reference/references we have to change.
- The author should delve deeper into some important characteristics of bread instead of irrelevant and unimportant indicators such as antioxidant activity.
This work has to be intended as a first evaluation about the potentiality of these formulations. Additional investigation about the macro and micronutrients contents (analytically determined) as well as the in loco implementation are mandatory further research activities to assess a wider sustainability of our proposals.
- The results in the table should be shown as mean±sd
Tables 2 and 5 include data showed as mean±sd. In Table 5, DS (digestible starch, RDS+SDS) and TS (total starch, DS+RS) don’t have standard deviations because they are calculated by the sum of other starch fractions as specified in brackets.
Tables 1 and 3 don’t require standard deviations.
For Table 4, it was impossible to retrieve the variations in raw materials costs in the local markets by local collaborators of the University of Parma.
Response to the reviewer is here attached

Round 2
Reviewer 1 Report
Dear all,
I am generally satisfied with the modifications which were done.
Kind regards
Reviewer 3 Report
All questions have been well addressed, and the revision can be accepted for publication.